# Equalized Robustness: Towards Sustainable Fairness Under Distributional Shifts

## Abstract

Increasing concerns have been raised on deep learning fairness in recent years. Existing fairness metrics and algorithms mainly focus on the discrimination of model performance across different groups on in-distribution data. It remains unclear whether the fairness achieved on in-distribution data can be generalized to data with unseen distribution shifts, which are commonly encountered in real-world applications. In this paper, we first propose a new fairness goal, termed Equalized Robustness (ER), to impose fair model robustness against unseen distribution shifts across majority and minority groups. ER measures robustness disparity by the maximum mean discrepancy (MMD) distance between the loss curvature distributions of two groups of data. We show that previous fairness learning algorithms designed for in-distribution fairness fail to meet the new robust fairness goal. We further propose a novel fairness learning algorithm, termed Curvature Matching (CUMA), to simultaneously achieve both traditional in-distribution fairness and our new robust fairness. CUMA debiases the model robustness by minimizing the MMD distance between loss curvature distributions of two groups. Experiments on three popular datasets show CUMA achieves superior fairness in robustness against distribution shifts, without more sacrifice on either overall accuracies or the in-distribution fairness.

## 1 Introduction

With the wide deployment of deep learning in modern business applications concerning individual lives and privacy, there naturally emerge concerns on machine learning fairness (Podesta et al., 2014; Muñoz et al., 2016; Smuha, 2019). Research efforts on various fairness evaluation metrics and corresponding enforcing methods have been carried out (Edwards & Storkey, 2016; Hardt et al., 2016; Du et al., 2020). Specifically, many such metrics require some form of "equalized model performance" across different groups on in-distribution data. Examples include Demographic parity (DP) (Edwards & Storkey, 2016), Equalized Opportunity (EOpp), and Equalized Odds (EO) (Hardt et al., 2016).

Unfortunately, when deployed for real-world applications, deep models commonly encounter data with unforeseeable distribution shifts (Hendrycks & Dietterich, 2019; Hendrycks et al., 2020; 2021). It has been shown that deep learning models can have drastically degraded performance (Hendrycks & Dietterich, 2019; Hendrycks et al., 2020; 2021; Taori et al., 2020) and show unreliable behaviors (Qiu et al., 2019; Yan et al., 2021) under unseen distribution shifts. Intuitively speaking, previous fairness learning algorithms aim to optimize the model to a local minimum where data from majority and minority groups have similar average loss values (and thus similar in-distribution performance). However, those algorithms do not take into consideration the the stability or "robustness" of their found fairness-aware minima. Taking object detection in a self-driving car for example, it might have been calibrated over high-quality clear images to be "fair" with different skin colors; however such fairness may severely break down when applied to data collected in adverse visual conditions, such as inclement weather, poor lighting, or other digital artifacts. Our experiments also find that previous state-of-the-art fairness algorithms would be jeopardized if distributional shifts are present in test data, as illustrated in Figure 1 (b). The above findings beg the following question:

*How to achieve practically sustainable fairness, e.g., even under unseen distribution shifts?*

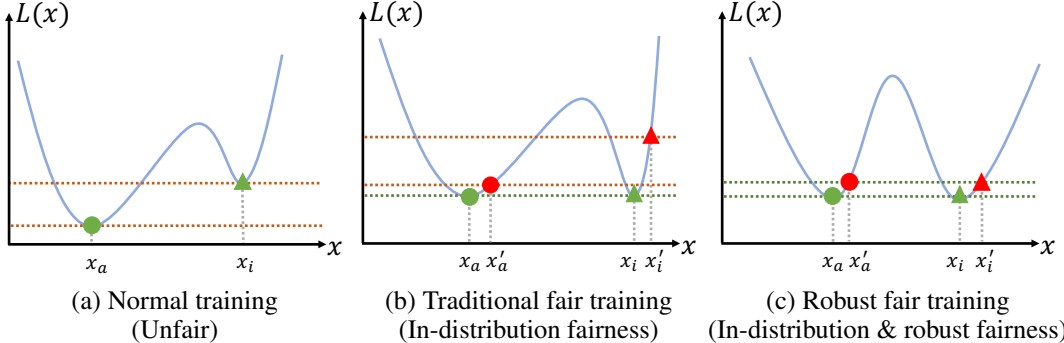

Figure 1: Illustrating the achieved fairness of normal training, traditional fair training and our proposed robust fair training algorithms. Horizontal and vertical axes represent input $x$ and corresponding loss value $\mathcal{L}(x)$, respectively. Solid blue curves show the loss landscapes. Circles denote majority data points ($x_a$ and $x'_a$), while triangles denote minority data points ($x_i$ and $x'_i$). Green points ($x_a$ and $x_i$) are in-distribution data while red ones ($x'_a$ and $x'_i$) are sampled from test sets with distribution shifts. (a) Normal training results in unfair models: minority group has worse performance (i.e., larger loss values). (b) Traditional fair training algorithms can achieve in-distribution fairness but not in a robust way: a small distribution shift can break the fairness due to loss curvature biases across different groups. In fact, such learned fair models can have almost the same large bias as the normally trained models when facing distribution shifts. (c) Our robust fair training algorithm can simultaneously achieve fairness both on in-distribution data and at distribution shifts, by matching both loss values and loss curvatures across different groups.

To answer that, we first propose a new fairness objective, termed **Equalized Robustness (ER)**, which aims to impose "equalized robustness" against unseen distribution shifts across the majority and minority groups, so that the learned fairness can sustain even with test data perturbed. ER explicitly considers a new dimension of fairness that is practically significant yet so far largely overlooked. In other words, ER assesses fairness on "out-of-distribution". Therefore it works as a *complement* instead of a *replacement* for previous fairness metrics, which focus on assessing the "in-distribution" fairness.

Previous research has shown that model robustness against input perturbation is highly correlated with loss curvature smoothness (Bartlett et al., 2017; Moosavi-Dezfooli et al., 2019; Weng et al., 2018). Our experiments also observed that, the local loss curvature of minority group is often larger than that of majority group, leading to the two group's robustness discrepancy against distribution shifts. To this end, we propose to empirically quantify the robustness discrepancy as the maximum mean discrepancy (MMD) (Gretton et al., 2012) distance between the local model smoothness distributions, for data samples from the majority and minority groups. We experimentally demonstrate that our new metric aligns well with model performance under real-world distribution shifts. On top of that, we further propose a new fair learning algorithm, termed **Curvature Matching (CUMA)**, to simultaneously achieve both traditional in-distribution fairness and ER. CUMA matches the local curvature distribution between data points from the two different groups, as illustrated in Figure 1 (c), by adding a curvature-matching regularizer that can be efficiently computed via a one-shot power iteration method. Our codes will be released upon acceptance.

Our contributions can be summarized as bellow:

- We propose *Equalized Robustness* (**ER**), a new fairness objective for machine learning models, to impose equalized model robustness against unforeseeable distributions shifts across majority and minority groups.

- We further propose a new fairness learning algorithm dubbed *Curvature Matching* (**CUMA**), which enforces ER during training by utilizing a one-shot power iteration method.

- Experiments show that CUMA achieves much more robust fairness against distribution shifts, without more sacrifice on either overall accuracies or the in-distribution fairness, compared with traditional in-distribution fair learning methods.

## 2 PRELIMINARIES

### 2.1 MACHINE LEARNING FAIRNESS

**Problem Setting and Metrics**    Machine learning fairness can be generally categorized into individual fairness and group fairness (Du et al., 2020). Individual fairness requires similar inputs to have similar predictions (Dwork et al., 2012). Compared with individual fairness, group fairness is a more popular setting and thus the focus of our paper. Given input data $X \in \mathbb{R}^n$ with sensitive attributes $A \in \{0, 1\}$ and their corresponding ground truth labels $Y \in \{0, 1\}$, group fairness requires a learned binary classifier $f(\cdot; \theta) : \mathbb{R}^n \to \{0, 1\}$ parameterized by $\theta$ to give equally accurate predictions (denoted as $\hat{Y} := f(X)$) on the two groups with $A = 0$ and $A = 1$. Multiple fairness criteria have been defined in this context. Demographic parity (DP) (Edwards & Storkey, 2016) requires identical ratio of positive predictions between two groups: $P(\hat{Y} = 1|A = 0) = P(\hat{Y} = 1|A = 1)$. Equalized Odds (EO) (Hardt et al., 2016) requires identical false positive rates (FPRs) and false negative rates (FNRs) between the two groups: $P(\hat{Y} \neq Y|A = 0, Y = y) = P(\hat{Y} \neq Y|A = 1, Y = y), \forall y \in \{0, 1\}$. Equalized Opportunity (EOpp) (Hardt et al., 2016) requires only equal FNRs between the groups: $P(\hat{Y} \neq Y|A = 0, Y = 0) = P(\hat{Y} \neq Y|A = 1, Y = 0)$. Based on these fairness criteria, quantified metrics are defined to measure fairness. Specifically, DP, EO and EOpp distances (Madras et al., 2018) are defined as follows:

$$\Delta_{DP} := |P(\hat{Y} = 1|A = 0) - P(\hat{Y} = 1|A = 1)| \tag{1}$$

$$\Delta_{EO} := \sum_{y \in \{0,1\}} |P(\hat{Y} \neq Y|A = 0, Y = y) - P(\hat{Y} \neq Y|A = 1, Y = y)| \tag{2}$$

$$\Delta_{EOpp} := |P(\hat{Y} \neq Y|A = 0, Y = 0) - P(\hat{Y} \neq Y|A = 1, Y = 0)| \tag{3}$$

MMD has been previously used to define fairness metric in (Quadrianto & Sharmanska, 2017) defines a more general fairness metric using MMD distance, and shows $\Delta_{DP}$, $\Delta_{EO}$ and $\Delta_{EOpp}$ to be spatial cases of their unified metric. All these metrics consider the in-distribution fairness, while our Equalized Generalizibility is the first fairness metric explicitly aware of robust generalization ability on unseen distributions.

**Bias Mitigation Methods**    Many methods have been proposed to mitigate model bias. Data pre-processing methods such as re-weighting (Kamiran & Calders, 2012) and data-transformation (Calmon et al., 2017) have been used to reduce discrimination before model training. In contrast, Hardt et al. (2016) and Zhao et al. (2017) propose post-processing methods to calibrate model predictions towards a desired fair distribution after model training. Instead of pre- or post-processing, researchers have explored to enhance fairness during training. For example, Madras et al. (2018) uses a adversarial training technique and shows the learned fair representations can transfer to unseen target tasks. The key technique, adversarial training (Edwards & Storkey, 2016), was designed for feature disentanglement on hidden representations such that sensitive (Edwards & Storkey, 2016) or domain-specific information (Ganin et al., 2016) will be removed while keeping other useful information for the target task. The hidden representations are typically the output of intermediate layers of neural networks (Ganin et al., 2016; Edwards & Storkey, 2016; Madras et al., 2018). Instead, methods, like adversarial debiasing (Zhang et al., 2018) and its simplified version (Wadsworth et al., 2018), directly apply the adversary on the output layer of the classifier, which also promotes the model fairness. Observing the unfairness due to ignoring the worst learning risk of specific samples, Hashimoto et al. (2018) proposes to use distributionally robust optimization which provably bounds the worst-case risk over groups. Creager et al. (2019) proposes a flexible fair representation learning framework based on VAE (Kingma & Welling, 2013), that can be easily adapted for different sensitive attribute settings during run-time. Sarhan et al. (2020) uses orthogonality constraints as a proxy for independence to disentangles the utility and sensitive representations. Martinez et al. (2020) formulates group fairness with multiple sensitive attributes as a multi-objective learning problem and proposes a simple optimization algorithm to find the Pareto optimality. Another line of research focuses on learning unbiased representations from biased ones (Bahng et al., 2020; Nam et al., 2020). Bahng et al. (2020) proposes a novel framework to learn unbiased representations by explicitly enforcing them to be different from a set of pre-defined biased representations. Nam et al. (2020) observes that data bias can be either benign or malicious, and removing malicious bias along can achieve fairness. Li & Vasconcelos (2019) jointly learns a data re-sampling weight distribution that penalizes easy samples and network parameters.

**Applications in Computer Vision** When many fairness metrics and debiasing algorithms are designed for general learning problems as aforementioned, there are a line of research and applications focusing on fairness-encouraged computer vision tasks. For instance, Buolamwini *et al.* (Buolamwini & Gebru, 2018) shows current commercial gender-recognition systems have substantial accuracy disparities among groups with different genders and skin colors. Wilson et al. (2019) observe that state-of-the-art segmentation models achieve better performance on pedestrians with lighter skin colors. In (Shankar et al., 2017; de Vries et al., 2019), it is found that the common geographical bias in public image databases can lead to strong performance disparities among images from locales with different income levels. Nagpal et al. (2019) reveal that the focus region of face-classification models depends on people's ages or races, which may explain the source of age- and race-biases of classifiers. On the awareness of the unfairness, many efforts have been devoted to mitigate such biases in computer vision tasks. Wang et al. (2019) shows the effectiveness of adversarial debiasing technique (Zhang et al., 2018) in fair image classification and activity recognition tasks. Beyond the supervised learning, FairFaceGAN (Hwang et al., 2020) is proposed to prevent undesired sensitive feature translation during image editing. Similar ideas have also been successfully applied to visual question answering (Park et al., 2020).

## 2.2 MODEL ROBUSTNESS AND SMOOTHNESS

Model generalization ability and robustness has been shown to be highly correlated with model smoothness (Moosavi-Dezfooli et al., 2019; Weng et al., 2018). Weng et al. (2018) and Guo et al. (2018) use local Lipschitz constant to estimate model robustness against small perturbations on inputs within a hyper-ball. Moosavi-Dezfooli et al. (2019) proposes to improve model robustness by adding a curvature constraint to encourage model smoothness. Miyato et al. (2018) approximates model local smoothness by the spectral norm of Hessian matrix, and improves model robustness against adversarial attacks by regularizing model smoothness.

## 3 EQUALIZED ROBUSTNESS: A NEW METRIC FOR FAIR GENERALIZATION AND ROBUSTNESS

Consider a binary classifier $f(\cdot; \theta)$ trained on two groups of data $X_1$ and $X_2$ respectively. Our goal is to define a metric to measure the gap of model robustness between the two groups. Formulating such a metric is highly non-trivial, with difficulties from mainly two aspects.

The first challenge is that we need to ensure fair generalization against *multiple unseen* distribution shifts that may encounter in real world applications. A trivial solution would be selecting a set of predefined distribution shifts and measuring the average performance gap (e.g., $\Delta_{EO}$) against them. However, this approach requires engineering overhead in handcrafting the predefined distribution shifts, and the predefined distribution shifts may not be representative enough to cover all unseen cases. Previous research (Miyato et al., 2018; Moosavi-Dezfooli et al., 2019; Guo et al., 2018; Weng et al., 2018) has shown both theoretically and empirically that deep model robustness scales with its model smoothness. Following (Miyato et al., 2018; Moosavi-Dezfooli et al., 2019), we use the spectral norm of Hessian matrix to approximate local smoothness as an indicator of model robustness. Specifically, given an input $x$, the Hessian matrix $H(x)$ is defined as the second-order gradient of $\mathcal{L}(x)$ with respect to input $x$: $H(x) = \nabla_x^2 \mathcal{L}(x)$. The approximated local curvature $\mathcal{C}(x)$ at point $x$ is thus defined as:

$$\mathcal{C}(x) = \sigma(H(x)), \tag{4}$$

where $\sigma(H)$ is the spectral norm of $H$: $\sigma(H) = \sup_{v:\|v\|_2=1} \|Hv\|_2$. Intuitively, $\mathcal{C}(x)$ measures the maximal directional curvature or change rate at $x$. Thus, smaller $\mathcal{C}(x)$ indicates better local smoothness around $x$ (Miyato et al., 2018; Moosavi-Dezfooli et al., 2019).

For the second difficulty, unlike previous fairness metrics where the target random variable[1] follows a Bernoulli distribution, the local curvature used in ER is a continuous random variable without a simple underlying distribution. The unknown distribution form makes it difficult to directly measure the difference between the curvature distributions by a parametric statistic test (e.g., t-test or KL divergence). To tackle this problem, we utilize maximum mean discrepancy (MMD) (Gretton et al.,

---

[1]Such as $Y = 1$ in DP and $Y \neq \hat{Y}$ in EO and EOpp. (See Section 2.1.)

2012) to do a two-sample test on $\mathcal{C}(X_1)$ and $\mathcal{C}(X_2)$. MMD is a distribution distance measure, agnostic to the exact distribution formulation and only based on the mean difference. Formally, our new fairness metric for equalized robustness is defined as follows:

**Our new fairness metric** $\Delta_{ER}$   Consider a machine learning model $f$ trained on two groups of data $X_1$ and $X_2$ respectively. Suppose $\mathcal{C}(X_1) \sim \mathcal{P}_1$ and $\mathcal{C}(X_2) \sim \mathcal{P}_2$, then the model's $\Delta_{ER}$ is defined as the squared maximum-mean-discrepancy (MMD) distance between $\mathcal{C}(X_1)$ and $\mathcal{C}(X_2)$:

$$\Delta_{ER} = \text{MMD}^2(\mathcal{P}_1, \mathcal{P}_2). \tag{5}$$

MMD is widely used to measure the distance between two high-dimensional distributions in deep learning (Li et al., 2015; 2017; Bińkowski et al., 2018). The MMD distance between two distributions $\mathcal{P}$ and $\mathcal{Q}$ is defined as

$$\text{MMD}^2(\mathcal{P}, \mathcal{Q}) = \|\mu_{\mathcal{P}} - \mu_{\mathcal{Q}}\|_{\mathcal{H}}^2 = \mathbb{E}_{\mathcal{P}}[k(X,X)] - 2\mathbb{E}_{\mathcal{P},\mathcal{Q}}[k(X,Y)] + \mathbb{E}_{\mathcal{Q}}[k(Y,Y)] \tag{6}$$

where $X \sim \mathcal{P}, Y \sim \mathcal{Q}$ and $k(\cdot, \cdot)$ is the kernel function. In practice, we use finite samples from $\mathcal{P}$ and $\mathcal{Q}$ to statistically estimate their MMD distance:

$$\text{MMD}^2(\mathcal{P}, \mathcal{Q}) = \frac{1}{M^2}\sum_{i=1}^{M}\sum_{i'=1}^{M}k(x_i, x_{i'}) - \frac{2}{MN}\sum_{i=1}^{M}\sum_{j=1}^{N}k(x_i, y_j) + \frac{1}{N^2}\sum_{j=1}^{N}\sum_{j'=1}^{N}k(y_j, y_{j'}) \tag{7}$$

where $\{x_i \sim \mathcal{P}\}_{i=1}^{M}$, $\{y_j \sim \mathcal{Q}\}_{j=1}^{N}$, and we use the mixed RBF kernel function $k(x,y) = \sum_{\sigma \in \mathbb{S}} e^{-\frac{\|x-y\|^2}{2\sigma^2}}$ with hyperparameter $\mathbb{S} = \{1, 2, 4, 8, 16\}$. Ablation studies on $\mathbb{S}$ values are conducted in Section 5.3.

# 4   CURVATURE MATCHING: FAIR MACHINE LEARNING TOWARDS EQUALIZED ROBUSTNESS

## 4.1   PRACTICAL CURVATURE APPROXIMATION

In order to achieve equalized robustness, one intuitive solution is to add $\Delta_{ER}$ (Eq. (5)) as an regularization term in the loss function during training phase. However, it is non-practical to precisely calculate the spectral norm (which is equal to the absolute value of dominant eigenvalue) of Hessian matrix in $\Delta_{ER}$. To solve this problem, we use a one-shot power iteration method (PIM) for practical approximation of $\mathcal{C}(x)$ during training. First we rewrite $\mathcal{C}(x)$ with the following form: $\mathcal{C}(x) = \sigma(H(x)) = \|H(x)v\|$, where $v$ is the dominant eigenvector with the maximal eigenvalue, which can be calculated by power iteration method. In practice, to increase training efficiency, we use a one-shot power iteration method. Specifically, we estimate the dominant eigenvector $v$ by the gradient direction: $\tilde{v} := \frac{\text{sign}(g)}{\|\text{sign}(g)\|} \approx v$, where $g = \nabla_x \mathcal{L}(x)$. This is because previous works have observed a large similarity between the dominant eigenvector and the gradient direction (Miyato et al., 2018; Moosavi-Dezfooli et al., 2019). We further approximate Hessian matrix by finite differentiation on gradients: $H(x)v \approx \frac{\nabla_x \mathcal{L}(x+hv) - \nabla_x \mathcal{L}(x)}{h}$ where $h$ is a small constant. As a result, the final approximation of curvature smoothness is

$$\mathcal{C}(x) \approx \tilde{\mathcal{C}}(x) := \frac{\|\nabla_x \mathcal{L}(x + h\tilde{v}) - \nabla_x \mathcal{L}(x)\|}{|h|}. \tag{8}$$

## 4.2   CURVATURE MATCHING

With the practical curvature approximation, now we can match the curvature distribution of the two groups by minimizing the MMD distance. Suppose $\tilde{\mathcal{C}}(X_1) \sim \mathcal{Q}_1$ and $\tilde{\mathcal{C}}(X_2) \sim \mathcal{Q}_2$, we define the curvature matching loss functions as:

$$\mathcal{L}_{cm} = \text{MMD}^2(\mathcal{Q}_1, \mathcal{Q}_2) \tag{9}$$

We add $\mathcal{L}_{cm}$ to the traditional adversarially fair training (Ganin et al., 2016; Madras et al., 2018) loss function as a regularizer, in order to attain both in-distribution fairness and fair robustness. As

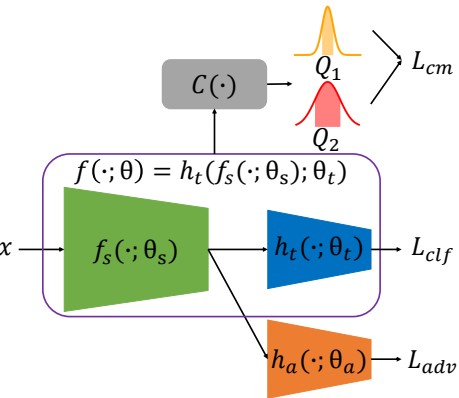

Figure 2: The overall framework of CUMA. $x$ is the input sample. $h_t$ is the utility head for the target task. $h_a$ is the adversarial head to predict sensitive attributes. $f_s$ is the shared backbone. $\mathcal{C}(\cdot)$ is the curvature estimation function, as defined in Eq. (4). $\mathcal{Q}_1$ and $\mathcal{Q}_2$ are local curvature distributions of majority and minority groups, respectively. $\mathcal{L}_{cm}$, $\mathcal{L}_{clf}$ and $\mathcal{L}_{adv}$ are three loss terms as defined in Eq. (9) and (11).

illustrated in Figure 2, our model follows the same "two-head" structure as traditional adversarial learning frameworks (Ganin et al., 2016; Madras et al., 2018), where $h_t$ is the utility head for the target task, $h_a$ is the adversarial head to predict sensitive attributes, and $f_s$ is the shared backbone.[2] Suppose for each sample $x_i$, the sensitive attribute is $a_i$ and the corresponding target label is $y_i$, then our overall optimization problem can be written as:

$$\min_{\theta_s, \theta_t} \max_{\theta_a} \mathcal{L} = \min_{\theta_s, \theta_t} \max_{\theta_a} (\mathcal{L}_{clf} - \alpha \mathcal{L}_{adv} + \gamma \mathcal{L}_{cm}) \tag{10}$$

where

$$\mathcal{L}_{clf} = \frac{1}{N} \sum_{i=1}^{N} \ell(h_t(f_s(x_i; \theta_s); \theta_t), y_i), \mathcal{L}_{adv} = \frac{1}{N} \sum_{i=1}^{N} \ell(h_a(f_s(x_i; \theta_s); \theta_a), a_i), \tag{11}$$

$\ell(\cdot, \cdot)$ is the cross-entropy loss function, $\alpha$ and $\gamma$ are trade-off hyperparameters, and $N$ is the number of training samples.

## 5 EXPERIMENTS

### 5.1 EXPERIMENTAL SETUP

**Datasets and pre-processing** Experiments are conducted on three datasets widely used to evaluate machine learning fairness: Communities and Crime (C&C) (Redmond & Baveja, 2002), Adult (Kohavi, 1996), and CelebA (Liu et al., 2015).[3] C&C dataset has 1,994 samples with neighborhood population statistics, where 1,500 are used for training and the rest for evaluation. The target task is to predict violent crime per capita, and we use "RacePctBlack" (percentage of black population in the neighborhood) and "FemalePctDiv" (divorce ratio of female in the neighborhood) as sensitive attributes. All features in C&C dataset are of continous values in $[0, 1]$. To fit in the fairness problem setting, we binarilize the target and sensitive attributes with the top-30% largest value as the threshold.[4] We also do data-whitening on C&C. Adult dataset has 48,842 samples with basic personal information such as education and occupation, where 30,000 are used for training and the rest for evaluation. The target task is to predict the person's annual income, and we use "gender" (male or female) as the sensitive attribute. The features in Adult dataset are of either continuous (e.g., age) or categorical (e.g. sex) values. We use one-hot encoding on the categorical features and then concatenate them with the continuous ones. We use data whitening on the concatenated features. CelebA has over 200,000 images of celebrity faces, with 40 attribute annotations. The target task is to predict the "attractiveness" attribute and the sensitive attributes to protect are "chubby" and "eyeglasses". We randomly select $45,000$ as training samples and $5,000$ as testing samples. All images are center-cropped and resized to $128 \times 128$, and pixel values are scaled to $[0, 1]$.

---

[2]Thus the binary classifier $f(\cdot; \theta) = h_t(f_s(\cdot; \theta_s); \theta_t)$, with $\theta = \theta_t \cup \theta_s$.

[3]Traditional image classification datasets (e.g., ImageNet) are not directly applicable since they lack fairness attribute labels.

[4]As a result P$[A = 0] = 30\%$ and P$[Y = 0] = 30\%$.

Table 1: Results on C&C dataset with "RacePctBlack" as the sensitive attribute. The best and second-best metrics are shown in bold and underlined, respectively.

| Method | Original Test Set | | | With Gaussian Noise | | With Uniform Noise | |
|---|---|---|---|---|---|---|---|
| | Accuracy (↑) | $\Delta_{EOpp}(\downarrow)$ | $\Delta_{EO}(\downarrow)$ | $\Delta_{ER}(\downarrow)$ | $\Delta_{EOpp}(\downarrow)$ | $\Delta_{EO}(\downarrow)$ | $\Delta_{EOpp}(\downarrow)$ | $\Delta_{EO}(\downarrow)$ |
| | | In-distribution fairness | | Robust fairness under distribution shifts | | | | |
| Normal | **89.05** | 38.52 | 63.22 | 46.16 | 35.43 | 60.13 | 39.51 | 64.21 |
| AdvDebias | 84.79 | 26.68 | 39.84 | 21.77 | 26.68 | 39.84 | 23.65 | 36.81 |
| LAFTR | 85.80 | 13.32 | 28.83 | 16.98 | 13.53 | 29.04 | 16.69 | 32.20 |
| CUMA | 85.20±1.70 | **12.71**±1.47 | **28.17**±1.70 | **7.59**±0.19 | **10.17**±0.89 | **28.69**±1.92 | **12.85**±2.98 | **27.11**±0.82 |

Table 2: Results on C&C dataset with "FemalePctDiv" as the sensitive attribute. The best and second-best metrics are shown in bold and underlined, respectively.

| Method | Original Test Set | | | With Gaussian Noise | | With Uniform Noise | |
|---|---|---|---|---|---|---|---|
| | Accuracy (↑) | $\Delta_{EOpp}(\downarrow)$ | $\Delta_{EO}(\downarrow)$ | $\Delta_{ER}(\downarrow)$ | $\Delta_{EOpp}(\downarrow)$ | $\Delta_{EO}(\downarrow)$ | $\Delta_{EOpp}(\downarrow)$ | $\Delta_{EO}(\downarrow)$ |
| | | In-distribution fairness | | Robust fairness under distribution shifts | | | | |
| Normal | **85.60** | 17.28 | 54.74 | 67.69 | 17.63 | 56.41 | 18.77 | 54.60 |
| AdvDebias | 83.57 | 12.80 | 38.73 | 37.17 | 12.80 | 38.73 | 11.38 | 37.15 |
| LAFTR | 83.16 | 11.73 | 27.83 | 28.15 | 11.73 | 29.30 | 11.38 | 30.11 |
| CUMA | 83.39±1.01 | **8.65**±0.59 | **27.57**±0.74 | **27.70**±1.04 | **8.71**±0.88 | **27.70**±1.04 | **9.63**±1.37 | **28.35**±1.73 |

**Models**   For C&C and Adult datasets, we use two-layer MLPs for $f_s$, $h_t$ and $h_a$. For CelebA dataset, we use ResNet18 as backbone, where the first three stages are used as $f_s$ and the last stage (together with the fully connected classification layer) is used as $h_t$. The auxiliary adversarial head $h_a$ has the same structure as $h_t$. Detailed model structures are described in Appx. A.

**Baseline Methods**   We compare CUMA with the following state-of-the-art in-distribution fairness algorithms. Adversarial debiasing (AdvDebias) (Zhang et al., 2018) is one of the most popular fair training algorithm based on adversarial training (Ganin et al., 2016). Madras et al. (2018) proposes a similar framework termed Learned Adversarially Fair and Transferable Representations (LAFTR), by replacing the cross-entropy loss used in (Zhang et al., 2018) with a group-normalized $\ell_1$ loss, which is shown to work better on highly unbalanced datasets. We also include normal (fairness-ignorant) training as a baseline.

**Evaluation Metric**   We use three different groups of evaluation metrics: the overall accuracy, in-distribution fairness metrics, and robust fairness metrics. We report the overall accuracy on all test samples in the original test sets. To measure in-distribution fairness, we use $\Delta_{EOpp}$ and $\Delta_{EO}$ on the original test sets. To measure robust fairness under distribution shifts, we use our newly proposed $\Delta_{ER}$ on the original test sets, and also $\Delta_{EOpp}$ and $\Delta_{EO}$ on a set of pre-defined real-world distribution shifts. We intend to show that $\Delta_{ER}$ calculated on the original test sets aligns well with robust fairness under real-world distribution shifts. See the following paragraph for the details in constructing distributional shifts.

**Distributional shifts**   On Adult and C&C datasets, we construct two distribution shifts by adding random Gaussian and uniform noises, respectively, to the test data. Specifically, following (Madras et al., 2018; Zhang et al., 2018), the categorical features in Adult and C&C datasets are first one-hot encoded and then whitened into float-value vectors, where noises are added. Both types of noises have mean $\mu = 0$ and has standard derivation $\sigma = 0.03$ . On CelebA dataset, following (Hendrycks & Dietterich, 2019), we construct two distribution shifts by adding random Gaussian (with mean $\mu = 0$ and standard derivation $\sigma = 0.08$) and impulse noise (with ratio $p = 0.03$), respectively. We report the fairness in robustness against other settings of distribution shifts in Appx. C.

**Implementation Details**   Unless further specified, we set the loss trade-off parameter $\alpha$ to 1 in all experiments by default. We use Adam optimizer (Kingma & Ba, 2014) with initial learning rate $10^{-3}$ and weight decay $10^{-5}$. The learning rate is gradually decreased to 0 by cosine annealing learning rate scheduler (Loshchilov & Hutter, 2016). On both Adult and C&C datasets, we train for 50 epochs from scratch for all methods. On CelebA dataser, we first normally train a model for 100 epochs, and then finetune it for 20 epochs using CUMA. For fair comparison, we train for 120 epochs on CelebA for all baseline methods. The constant $h$ in Eq. (8) is set to 1 by default. For more implementation details, please check Appx. A.

Table 3: Results on Adult dataset with "Sex" as the sensitive attribute. The best and second-best metrics are shown in bold and underlined, respectively.

| Method | Original Test Set | | | With Gaussian Noise | | With Uniform Noise | |
|---|---|---|---|---|---|---|---|
| | Accuracy ($\uparrow$) | $\Delta_{EOpp}(\downarrow)$ | $\Delta_{EO}$ ($\downarrow$) | $\Delta_{ER}$ ($\downarrow$) | $\Delta_{EOpp}(\downarrow)$ | $\Delta_{EO}$ ($\downarrow$) | $\Delta_{EOpp}(\downarrow)$ | $\Delta_{EO}$ ($\downarrow$) |
| | | In-distribution fairness | | Robust fairness under distribution shifts | | | | |
| Normal | **86.11** | 6.65 | 15.45 | 34.25 | 6.66 | 15.01 | 6.87 | 15.72 |
| AdvDebias | 85.17 | 5.12 | 5.92 | 16.78 | 5.10 | 5.95 | 5.77 | 7.29 |
| LAFTR | 85.97 | 6.28 | 11.96 | 25.38 | 6.22 | 12.08 | 6.45 | 12.06 |
| CUMA | 85.30±0.73 | **4.83**±0.24 | **4.77**±0.34 | **5.59**±0.28 | **4.74**±0.32 | **4.81**±0.51 | **5.43**±0.19 | **6.87**±0.31 |

Table 4: Results on CelebA dataset with "Chubby" as the sensitive attribute. The best and second-best metrics are shown in bold and underlined, respectively.

| Method | Original Test Set | | | With Gaussian Noise | | With Impulse Noise | |
|---|---|---|---|---|---|---|---|
| | Accuracy ($\uparrow$) | $\Delta_{EOpp}(\downarrow)$ | $\Delta_{EO}$ ($\downarrow$) | $\Delta_{ER}$ ($\downarrow$) | $\Delta_{EOpp}(\downarrow)$ | $\Delta_{EO}$ ($\downarrow$) | $\Delta_{EOpp}(\downarrow)$ | $\Delta_{EO}$ ($\downarrow$) |
| | | In-distribution fairness | | Robust fairness under distribution shifts | | | | |
| Normal | **91.25** | 38.45 | 42.56 | 59.34 | 39.16 | 43.90 | 39.76 | 44.51 |
| AdvDebias | 90.48 | **26.41** | 29.73 | 42.65 | 28.95 | 35.46 | 29.73 | 36.48 |
| LAFTR | 89.92 | 26.54 | **29.10** | 39.16 | 27.94 | 34.60 | 28.96 | 35.12 |
| CUMA | 89.97±0.38 | 27.19±0.75 | 30.26±0.95 | **23.23**±0.39 | **27.62**±0.85 | **31.49**±1.28 | **27.97**±0.48 | **31.74**±1.14 |

## 5.2 MAIN RESULTS

Experimental results on three datasets with different sensitive attributes are shown in Tables 3-5, where we compare CUMA with the baseline methods on three different groups of metrics as discussed in Section 5.1. "Normal" means standard training without any fairness regularization. All numbers are shown as percentages. Many intriguing findings can be concluded from the results.

First, we see that previous state-of-the-art fairness learning algorithms would be jeopardized if distributional shifts are present in test data. For example, on C&C dataset with "RacePctBlack" as sensitive attribute (Table 1), LAFTR achieves $\Delta_{EO} = 28.83\%$ on in-distribution test set, while that number is increased to $32.20\%$ on the test set perturbed with uniform random noise. Similarly, for AdvDebias, it achieves $\Delta_{EO} = 29.73\%$ on the original CelebA test set with "chubby" as the sensitive attribute (Table 4), while that number is increased to $35.46\%$ and $36.48\%$ on test sets perturbed with Gaussian and impulse noises, respectively.

Second, we see that CUMA achieves the best robust fairness under distribution shifts on all three benchmark datasets with different sensitive attribute settings, while maintaining similar in-distribution fairness and overall accuracy. For example, on C&C dataset with "RacePctBlack" as the sensitive attribute (Table 1), CUMA achieves $2.73\%$ and $4.82\%$ less $\Delta_{EO}$ than the second-best performer (LAFTR) under distribution shifts by additive Gaussain and uniform noises, respectively. Moreover, for the same experiment setting, although CUMA and LAFTR achieve almost identical in-distribution fairness (the difference between their $\Delta_{EO}$ on original test set is within $0.5\%$), CUMA keeps (and even increases) the fairness under distribution shifts (e.g., $1.33\%$ smaller $\Delta_{EO}$ under uniform noises), while the fairness achieved by LAFTR is jeopardized under both types of distribution shifts (e.g., $3.37\%$ larger $\Delta_{EO}$ under uniform noises). Similarly, on CelebA dataset with "Chubby" as the sensitive attribute, LAFTR has even slightly better in-distribution fairness than CUMA. However, when the test sets have distribution shifts, the fairness achieved by LAFTR is jeopardized (with $5.50\%$ and $6.02\%$ more $\Delta_{EO}$ under Gaussian and uniform noises, respectively), while CUMA keeps its fairness and achieves better fairness under distribution shifts (e.g., $2.50\%$ and $3.17\%$ less $\Delta_{EO}$ compared with LAFTR.).

Third, for all three datasets, the $\Delta_{ER}$ calculated on the original test set highly correlates with traditional fairness metrics (e.g., $\Delta_{EOpp}, \Delta_{EO}$) calculated on the perturbed test sets: the smaller $\Delta_{ER}$ on the in-distribution test set, the smaller $\Delta_{EO}$ on perturbed test sets. This shows that our new metric $\Delta_{ER}$ aligns well with robust fairness under real-world distribution shifts, and validates the rationality of using it as an indicator of model robustness discrimination.

More experimental results are shown in Appx. B (trade-off curves between fairness and accuracy) and Appx. C (results on other settings of distributional shifts).

Table 5: Results on CelebA dataset with "Eyeglasses" as the sensitive attribute. The best and second-best metrics are shown in bold and underlined, respectively.

| Method | Original Test Set | | | With Gaussian Noise | | With Impulse Noise | |
| | Accuracy (↑) | $\Delta_{EOpp}(\downarrow)$ | $\Delta_{EO}(\downarrow)$ | $\Delta_{ER}(\downarrow)$ | $\Delta_{EOpp}(\downarrow)$ | $\Delta_{EO}(\downarrow)$ | $\Delta_{EOpp}(\downarrow)$ | $\Delta_{EO}(\downarrow)$ |
| | | In-distribution fairness | | Robust fairness under distribution shifts | | | | |
|---|---|---|---|---|---|---|---|---|
| Normal | **90.52** | 36.40 | 43.96 | 54.38 | 35.62 | 42.91 | 37.92 | 45.63 |
| AdvDebias | 88.65 | **23.15** | **32.56** | 41.06 | **25.70** | 36.41 | 23.92 | 33.46 |
| LAFTR | 89.72 | 24.90 | 35.48 | 42.93 | 26.12 | 37.94 | 24.52 | 34.10 |
| CUMA | 89.10±0.13 | 24.16±0.40 | 33.39±0.22 | **32.56**±0.41 | 25.76±0.50 | **34.77**±0.47 | **22.61**±0.06 | **31.68**±0.15 |

## 5.3 ABLATION STUDY

**Ablation Study on $\Delta_{ER}$** In this section, we study how well can the $\Delta_{ER}$ predict the robust fairness and the sensitivity of $\Delta_{ER}$ with respect to $\mathbb{S}$ (the sampling set for $\sigma$ in the mixed RBF kernel function, as described in Section 3). A small $\sigma$ will make the $\Delta_{ER}$ more sensitive to the difference between the two sample set, which could be caused by either the true discrepancy of distributions or the different noise introduced by sampling. In contrast, a larger ones may under-estimate the discrepancy. Thus, a proper $\mathbb{S}$ should include a wide range of $\sigma$ to avoid the domination of either large or small values. In this paper, we choose a geometric sequence with 2 as the base, i.e., $\mathbb{S} = \{1, 2, 4, 8, 16\}$. Furthermore, we compare $\Delta_{ER}$ values under three different sets: $\mathbb{S}_1 = \{0.25, 0.5, 1, 2, 4\}$, $\mathbb{S}_2 = \{1, 2, 4, 8, 16\}$ (the default $\mathbb{S}$ as defined in Section 3), and $\mathbb{S}_3 = \{4, 8, 16, 32, 64\}$. Results are shown in

Table 6: $\Delta_{ER}$ values with different mixed RBF kernel scale parameter set $\mathbb{S}$. Results are reported on C&C dataset with "RacePctBlack" as the sensitive attribute. Models are trained by CUMA with different $\gamma$ values.

| | $\Delta_{ER}$ on Original Test Set | | | $\Delta_{EO}$ on Test Set |
| | $\mathbb{S} = \mathbb{S}_1$ | $\mathbb{S} = \mathbb{S}_2$ | $\mathbb{S} = \mathbb{S}_3$ | with Uniform Noise |
|---|---|---|---|---|
| $\gamma = 0.1$ | 12.72 | 13.52 | 11.06 | 31.09 |
| $\gamma = 1$ | 8.56 | 7.61 | 4.22 | 27.02 |
| $\gamma = 10$ | 8.40 | 7.24 | 4.02 | 26.98 |

Table 6. As in Section 5.2, we empirically evaluate the robust fairness by $\Delta_{EO}$ on the test set corrupted by uniform noise. From the results, we observe that with all three different $\mathbb{S}$ settings, $\Delta_{ER}$ aligns well with the model fairness under distribution shifts ($\Delta_{EO}$ under uniform noise).

**Ablation Study on CUMA** In this section, we check the sensitivity of CUMA with respect to its hyper-parameters: the loss trade-off parameters $\alpha$ and $\gamma$ in Eq. (10) and $h$ in Eq. (8). Results are shown in Table 7. When fixing $\gamma = 1$, $\Delta_{ER}$ peeks at around $\alpha = 1$, so we use

Table 7: Ablation study results on the loss trade-off parameters $\alpha$ and $\gamma$ in the CUMA algorithm. Results are reported on C&C dataset with "RacePctBlack" as the sensitive attribute.

| | $\alpha$ | | | $\gamma$ | | | $h$ | |
| | 0.1 | 1 | 10 | 0.1 | 1 | 10 | 0.1 | 1 |
|---|---|---|---|---|---|---|---|---|
| Accuracy | 86.94 | 85.40 | 83.75 | 85.19 | 85.40 | 84.79 | 85.32 | 85.40 |
| $\Delta_{EO}$ | 59.74 | 28.35 | 32.68 | 38.85 | 28.35 | 27.99 | 29.15 | 28.35 |
| $\Delta_{ER}$ | 42.50 | 7.61 | 18.56 | 13.52 | 7.61 | 7.24 | 7.53 | 7.61 |

it as the default $\alpha$ value. When fixing $\alpha = 1$, the best trade-off between overall accuracy and robust fairness is achieved at round $\gamma = 1$, which we use as the default $\gamma$. Varying the value of $h$ hardly affects the performance of CUMA.

## 6 CONCLUSION

In this paper, we first propose a new fairness goal, termed Equalized Robustness (ER), to impose fair model robustness against unseen distribution shifts across different data groups. We further propose a novel fairness learning algorithm, termed Curvature Matching (CUMA), to simultaneously achieve both traditional in-distribution fairness and our new robust fairness. Experiments show CUMA achieves superior fairness in robustness against distribution shifts, without more sacrifice on either overall accuracies or the in-distribution fairness compared with traditional in-distribution fair learning methods. As a pioneer work, the new concept of ER proposed in this paper aims to measure a new dimension of fairness that is practically significant yet so far largely overlooked: ER assesses "out-of-distribution" fairness while previous metrics focus on "in-distribution" fairness. Therefor, ER works as a complement instead of a replacement for previous fairness metrics. We hope our work can open up more discussions on how to evaluate model fairness in a more complete spectrum.

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

## A    More Implementation Details

On C&C and Adult datsets, suppose the input feature dimension is $d$, then the dimensions of hidden layers in $f_s$ and $h_t$ are $d \rightarrow 100 \rightarrow 64$ and $64 \rightarrow 32 \rightarrow 2$, respectively. $h_a$ has identical model structure with $h_t$. For all three sub-networks, ReLU activation function and dropout layer with $0.25$ dropout ratio are applied between the two fully connected layers. On CelebA dataset, we use the ResNet18 as backbone. The input feature size of $h_t$ and $h_a$ is $8 \times 8 \times 256$ (with channel-last layout).

## B    Trade-off Curves between Fairness and Accuracy

For CUMA and both baseline methods, we can obtain different trade-offs between fairness and accuracy by setting the loss function weights (e.g., $\alpha$ and $\gamma$) to different values. For example, the larger $\alpha$, the better fairness and the worse accuracy. Such trade-off curves between fairness and accuracy of different methods are shown in Figure 3. The closer the curve to the top-left corner (i.e., with larger accuracy and smaller $\Delta_{EO}$), the better Pareto frontier is achieved. As we can see, our method achieves the best Pareto frontiers for both in-distribution fairness (left panel) and robust fairness under distribution shifts (middle and right panel).

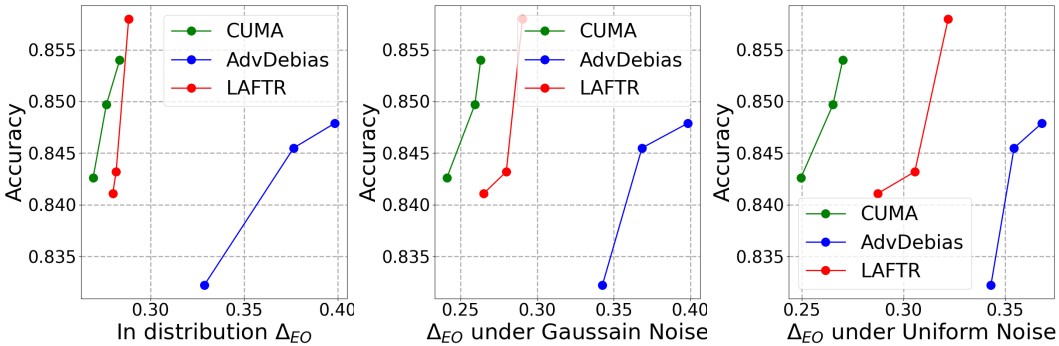

Figure 3: Trade-off curves between fairness and accuracy of different methods. Results are reported on C&C dataset with "RacePctBlack" as the sensitive attribute.

## C    Results on Other Settings of Distributional Shifts

Table 8: Results on C&C dataset with "RacePctBlack" as the sensitive attribute. The best and second-best metrics are shown in bold and underlined, respectively.

| Method | Accuracy ($\uparrow$) | Original Test Set | | | With Gaussian Noise | | With Uniform Noise | |
|---|---|---|---|---|---|---|---|---|
| | | $\Delta_{EOpp}(\downarrow)$ | $\Delta_{EO}$ ($\downarrow$) | $\Delta_{ER}$ ($\downarrow$) | $\Delta_{EOpp}(\downarrow)$ | $\Delta_{EO}$ ($\downarrow$) | $\Delta_{EOpp}(\downarrow)$ | $\Delta_{EO}$ ($\downarrow$) |
| | | *In-distribution fairness* | | | *Robust fairness under distribution shifts* | | | |
| Normal | **89.05** | 38.52 | 63.22 | 46.16 | 36.71 | 61.54 | 40.22 | 63.17 |
| AdvDebias | 84.79 | 26.68 | 39.84 | 21.77 | 28.61 | 37.02 | 22.84 | 37.41 |
| LAFTR | 85.80 | 13.32 | 28.83 | 16.98 | 13.96 | 31.25 | 16.58 | 33.42 |
| CUMA | 85.40 | **12.52** | **28.35** | **7.61** | **11.76** | **27.15** | **12.80** | **27.41** |

Table 9: Results on C&C dataset with "FemalePctDiv" as the sensitive attribute. The best and second-best metrics are shown in bold and underlined, respectively.

| Method | Accuracy ($\uparrow$) | Original Test Set | | | With Gaussian Noise | | With Uniform Noise | |
|---|---|---|---|---|---|---|---|---|
| | | $\Delta_{EOpp}(\downarrow)$ | $\Delta_{EO}$ ($\downarrow$) | $\Delta_{ER}$ ($\downarrow$) | $\Delta_{EOpp}(\downarrow)$ | $\Delta_{EO}$ ($\downarrow$) | $\Delta_{EOpp}(\downarrow)$ | $\Delta_{EO}$ ($\downarrow$) |
| | | *In-distribution fairness* | | | *Robust fairness under distribution shifts* | | | |
| Normal | **85.60** | 17.28 | 54.74 | 67.69 | 18.52 | 57.64 | 20.25 | 55.52 |
| AdvDebias | 83.57 | 12.80 | 38.73 | 37.17 | 14.90 | 39.60 | 12.58 | 35.26 |
| LAFTR | 83.16 | 11.73 | 27.83 | 28.15 | 13.12 | 30.21 | 12.41 | 31.52 |
| CUMA | 83.37 | **8.90** | **27.79** | **23.13** | **9.12** | **28.74** | **9.96** | **29.23** |

In this section, we show that the conclusions drawn in Section 5.2 hold under different settings of distributional shifts. Specifically, we consider a new noise setting with mean $\mu = 0$ and standard

derivation $\sigma = 0.06$ (other than the mean $\mu = 0$ and standard derivation $\sigma = 0.03$ evaluated in the main text) for both random Gaussian and uniform noises. The results under these new distributional shifts on C&C dataset are shown in Tables 8 and 9.

