# OpenReview forum: "Equalized Robustness: Towards Sustainable Fairness Under Distributional Shifts"
_ICLR.cc/2022/Conference — ICLR 2022 Submitted_

### Official Review · Reviewer_xUAG · 2021-10-28

**Correctness:** 3
**Technical Novelty And Significance:** 4
**Empirical Novelty And Significance:** 3
**Recommendation:** 8
**Confidence:** 3

**Main Review:**

+ The paper is very well-written and well-motivated. The paper provides a clear motivating example in the context of self-driving cars: a model may be fair with respect to skin tone for optimal light conditions, but disparities may arise when there is rain, poor lighting, etc. Figure 1 also provides a nice visual illustration of the key concepts.
+ The paper is very accessible. The notation is clear and the authors provide intuition about the key concepts.

- Does matching curvatures only guarantee that the groups will have similar level robustness which may be low or high robustness? Or does it specifically encourage high robustness?
- Would be helpful to be more specific about distributional shifts. It appears these are covariate shifts—i.e. shifts in the distribution of features/predictors, but assuming there is no shift in p(y |x).
- Do we expect the simulated distribution shifts (Gaussian and uniform noise) to be representative of real-world distribution shifts? Is it possible to empirically analyze a real world distribution shift?
- It is important to acknowledge the problems of a task like predicting "attractiveness".

Minor
- Would be useful to discuss the choice of kernel for MMD and what that may impact
- Empirical results should include uncertainty estimates
- I found the explanation of how hyper parameter S impacts the MMD to be confusing. I did not follow the sentences starting with: “A small sigma will make ….” Does small refer to the value of the spectral norm or to the number of sigmas in the sampled set?
- Would be helpful to include dimensions in introduction of the setting in section 3. E.g. x is a p-dimensional, Hessian is p x p matrix, etc



**Summary Of The Paper:**

This paper proposes a new fairness metric, Equalized Robustness (ER), to assess model robustness for sensitive subgroups. ER measures the maximum mean discrepancy (MMD) between the loss curvature of the two sensitive groups. The paper proposes using gradient direction to approximate the spectral norm and using finite differences to approximate the Hessian. The MMD is calculated using the RBF kernel.

 The paper shows that existing in-distribution fairness promoting methods do not achieve parity with respect to the new metric. A new method, curvature matching (CUMA), is proposed to achieve both in-distribution and robust fairness. CUMA adds the MMD distance as a regularizer to the standard adversarially fair learning approach (which includes a cross entropy loss term and an adversarially fair loss term for in-distribution fairness).

The paper performs ablation studies for the hyper parameter used in the RBF used to compute MMD.






**Summary Of The Review:**

The ability for fairness properties to generalize is an important problem tackled by this paper. This paper makes a couple novel contributions: 1) a fairness metric that captures robustness to distributional shifts; 2)  learning method to achieve this notion of “out-of-distribution” fairness as well as standard in-distribution fairness. The main technical areas of improvement are 1) include analysis on how well this method works for real-world (as opposed to simulated) distributional shifts and 2) provide error bars to support the empirical claims. I would also encourage the authors to exercise more care/caution around a task like predicting "attractiveness".

---

> ### Author Response · Authors · 2021-11-18
> **Response to Reviewer xUAG**
>
> Thanks for acknowledging the contributions of our work and providing the great suggestions. We have carefully answered your questions. Please check below.
>
> Q: Does matching curvatures only guarantee that the groups will have similar level robustness which may be low or high robustness?
>
> A: Yes, matching curvatures explicitly encourages similar levels of robustness, no matter high or low. Empirically, we find that the overall robustness of CUMA is similar to that of a normal training model. In other words, the overall robustness is not specifically improved nor decreased; it just gets more fair between the two groups, which is our goal.  For example, on C&C dataset, normal training has 81.36% accuracy under Gaussian noise. Our method has a slimiar 81.65% accuracy under the same settings.
>
> Q: Be more specific about distributional shifts.
>
> A: Yes you are right. We mainly focus on covariate shifts, and we will update in the final version.
>
>
> Q:  Analyse a real world distribution shift.
>
> A: We agree it is necessary to evaluate the method on real distributional shifts. We test the models trained on the Adult dataset (US Census data collected before 1996) on the 2014 subset of  Folktables dataset (US Census data collected in 2014). This is a practical setting to evaluate real-world distributional shifts caused by time. We use “sex” as the sensitive attribute and compare the ΔEOpp and ΔEO (the smaller the better) of different methods on Folktables (2014) subset:
>
> Normal: ΔEOpp=12.64%, ΔEO=22.58%
>
> AdvDebias: ΔEOpp=10.84%, ΔEO=13.65%
>
> LAFTR: ΔEOpp=9.84%, ΔEO=12.10%
>
> CUMA: ΔEOpp=6.21%, ΔEO=7.89%
>
> The results show that CUMA outperforms all baseline methods on fairness under real-world distributional shifts.
>
>
> Q: Acknowledge the problems of a task like predicting "attractiveness"
>
> A: Yes, we acknowledge that attractiveness prediction has ethical problems and will point that out in the paper.
>
>
> Q: The choice of kernel for MMD
>
> A: Currently we use the RBF kernel following (Li et al., 2015). We have also tried a polynomial kernel. We find the RBF kernel generally achieves better performance. For example, on Adult dataset with “Chubby” as the sensitive attribute and impulse noise, the polynomial kernel achieves ∆EOpp=28.62% and ∆EO=32.75%. In the same setting, the RBF kernel achieves  ∆EOpp=27.58% and ∆EO=31.95%.
>
> Q: Add uncertainty estimates
>
> A: We run CUMA using three different random seeds and report the error bars in the updated Table 1. As we can see, the improvements made by CUMA are statistically stable.
>
>
> Q: How does hyper parameter S impact the MMD?
>
> A: “Small” refers to the value of sigmas in the sampled set. The message we want to convey is that small sigmas make the RBF kernel over-sensitive to sampling noise, while large sigmas make it under-sensitive to real distribution gaps. As a result, we need to include a range of different sigmas for reliable estimation of the distance between two distributions, following (Li et al., 2017). More detailed explanations on how to choose sigma can be found in (Gretton et al., 2012).
>
> Q: Include dimensions for the notations in Section 3.
>
> A: Thanks for your suggestion and we will do it.

---

### Official Review · Reviewer_bgyC · 2021-11-01

**Correctness:** 3
**Technical Novelty And Significance:** 2
**Empirical Novelty And Significance:** 1
**Recommendation:** 3
**Confidence:** 4

**Main Review:**

Points in favor
+ Fairness is a hot and timely topic.
+ Selection of problem domain, i.e., fairness under distribution shifts.


Points against
- The authors overstate several of their contributions while the actual contribution is unknown. This paper does not propose a new fairness goal and the first fairness metric under distribution shifts. Rather it extends equalized model performance to distribution shift settings. Similarity, the 'new' learning algorithm is basically a combination of existing techniques such as Hessian matrix and adversarial learning frameworks. The authors should tone down these claims. It is also a stretch to argue that the proposed method does not achieve "much more robust fairness" and is insignificant instead. The statistic test is therefore suggested to verify the importance of these results.

- The paper does not discuss and compare proper state of the art. The related work discusses pre-, in-, and post-processing approaches while studies focus on fairness under distribution shifts are not cited and compared. To give concrete examples, fairness drift are considered in "FARF: A Fair and Adaptive Random Forests Classifier" as well as reference therein. I urge the authors to compare their method with recent works that consider fairness drift and discuss at least incomparability otherwise.

- Format violations such as the deceased margins before and after equations.

**Summary Of The Paper:**

This paper studies fairness under distribution shifts. The model claims to be fair in robustness against distribution shifts with several experiments being conducted.

**Summary Of The Review:**

Limited contribution, inappropriate related work and format violations.

---

> ### Author Response · Authors · 2021-11-18
> **Response to Reviewer bgyC**
>
> Thanks for your comments. With all respect, we find some of your comments are confusing due to conciseness and also inconsistent with all the other three reviewers. We have tried to address your questions carefully. We would really appreciate it if you could provide more detailed explanations and concreate examples to support your assessments, so that we can more properly address your concerns.
>
> Q: This paper does not propose a new fairness goal and the first fairness metric under distribution shifts. Rather it extends equalized model performance to distribution shift settings.
>
> A: We respectfully argue that our equalized robustness (ER) goal is not a naive extension of previous fairness goals.
>
> Previous fairness goals require equalized performance on in-distribution accuracy. Our new fairness goal (equalized robustness) requires equalized performance under multiple unseen distributional shifts. This is not a naive extension, as we described in Section 3. Specifically, we described two challenges to define the new metric for equalized robustness.
>
> The first is that we need to ensure fair generalization against multiple unseen distribution shifts that may be encountered in real world applications. To solve this challenge, we use curvature as a metric for general robustness against small input perturbations.
>
> The second is that, unlike previous fairness metrics where the target random variable follows a Bernoulli distribution, the local curvature used in ER is a continuous random variable without a simple underlying distribution. To solve this problem, we use MMD to measure the gap between two curvature distributions.
>
> Our final metric ∆ER in Eq. (5) is obviously different from the previous fairness metrics in Eq. (1)-(3).
>
> All other three reviewers acknowledged the novelty of our new fairness goal:
>
> Reviewer ebvu: “The paper proposes a new objective: Equalized Robustness (ER), that imposes equalized model robustness against distributions shifts across majority and minority.”
>
> Reviewer cgsp: “The idea is new and interesting.”
>
> Reviewer xUAG: “This paper proposes a new fairness metric, Equalized Robustness (ER), to assess model robustness for sensitive subgroups.”
>
> Q: The 'new' learning algorithm is basically a combination of existing techniques such as Hessian matrix and adversarial learning frameworks.
>
> A: We respectfully disagree with your assessment by pointing out: (i) we are the first to rigorously formulate and investigate the important problem of equalized robustness; (ii) our learning algorithm CUMA is a novel and organic combination of ideas from both robustness and fairness communities, customized for this new important problem; HOWEVER, CUMA is noticeably different from traditional robust learning or fairness algorithms. Details follow.
>
> ​​
> On one hand, the optimization goals of traditional robust learning algorithms are different from ours. Traditional robust learning papers such as (Moosavi-Dezfooli et al., 2019) minimize the mean curvature over all samples. However, in unbalanced datasets with majority and minority groups, such traditional robust learning methods would result in unbalanced robustness: the majority group can have much better robustness than the minority group. In contrast, our method aims to achieve fair robustness, by minimizing the discrepancy between the curvature distributions on two groups (majority and minority) of data - an untackled angle by any existing robust learning paper.
>
> On the other hand, as explained in the last paragraph on page 4, our “out-of-distribution” fairness learning is also different from traditional “in-distribution” fairness learning. The target random variable in traditional “in-distribution” fairness learning follows a Bernoulli distribution and the distance between two Bernoulli distributions can be directly calculated.
> In contrast, the local curvature used in our  “out-of-distribution” fairness learning is a continuous random variable without a simple underlying distribution. As a result, we cannot directly measure the “out-of-distribution” fairness using the divergence between the two distributions as done in traditional “in-distribution” fairness learning. Instead, we optimize the MMD between two curvature distributions to achieve equalized robustness.
>
> We hence respectfully argue: It would be unfair to rush to deny our novelty, solely because our algorithm is extended from relevant ideas in the past (perhaps true for many influential ML papers), especially given our extension is substantial. We hope the above clarifications make it more clear where our true novelty lies. We also appreciate reviewer ebvu for liking our “new objective and algorithm” and reviewer xUAG for endorsing us making “a couple of novel contributions” in both the fairness metric and learning algorithm.

---

> > ### Comment · Reviewer_bgyC · 2021-11-20
> > **Added after author response**
> >
> > Thank authors for taking time to read my reviews. I have read the other reviews and the author's response. The authors address some of my concerns in their response, such as format violations. However, the discussion and comparison with state of the art remains insufficient, particularly unforeseen distributional shift is still a type of distributional shift depending on the type of concept drift presents. Together with marginal improvements especially considering the presence of noise and the limited novelty, my score remains the same.

---

> > > ### Author Response · Authors · 2021-11-20
> > > **We are confused by vague points. More details to ground your assessment will be welcome.**
> > >
> > > Thanks so much for your swift reply, but we are genuinely confused by your assessment/accusation and do not find all your points well grounded.
> > >
> > > At this moment, we feel your further question to still result from misunderstanding our response. Perhaps, that is because we are not given sufficient details in either your original review or your latest response (both are relatively concise), so we will appreciate any more detailed clarifications.
> > >
> > > (1) Comparing with FARF:
> > >
> > > It seems to us you insist that the online learning paper named FARF can be applied in our problem setting and should be compared with. We did not see how this could be valid.
> > >
> > > As an online learning algorithm, FARF adapts the model to be fair on the current data distribution, from which training data is sampled for FARF (seen domain shifts). However, our work aims to generalize fairness learned on current distribution to unknown target distributions (unseen domain shifts). In our setting, the algorithm can not access any training data from the unknown target distributions. How can you apply FARF to adapt the model to an unknown target distribution, from which you cannot sample any training data? Could you reply and educate us with your concrete thoughts?
> > >
> > > Also, you vaguely accused "limited novelty", which seems to also arise from your strong understanding that our task setting is similar to FARF etc. Respectfully yet firmly, we believe you understand wrong as aforementioned explained, but we are very happy to discuss further if you ask with further details, more references etc.
> > >
> > > (2) "Marginal Improvement" and "Noise and uncertainty" in results
> > >
> > > It is again, too vague for us to address any further. The main results of our method are **all averaged over three random runs**, then outperforming all baselines by a considerable margin (below we show the mean and standard deviation of the improvements):
> > > - In Table 1, with uniform noise, CUMA outperforms the strongest baseline (LAFTR) by 5.09$\pm$0.82% on ∆EO.
> > > - In Table 2, with Gaussian noise, CUMA outperforms the strongest baseline (LAFTR) by 3.02$\pm$0.88% on ∆EOpp.
> > > - In Table 4, with Impulse noise, CUMA outperforms the strongest baseline (LAFTR) by 3.38$\pm$1.14% on ∆EO. With Gaussian Noise, CUMA outperforms the strongest baseline (LAFTR) by 3.11$\pm$1.28% on ∆EO.
> > > - In Table 5, with Impulse noise, CUMA outperforms the strongest baseline (AdvDebias) by 1.78$\pm$0.15% on ∆EO. With Gaussian Noise, CUMA outperforms the strongest baseline (AdvDebias) by 1.64$\pm$0.47% on ∆EO.
> > >
> > > This shows the improvements made by our algorithm are **large AND statistically significant**. We would like to invite you to give more supporting evidence on your claims "Together with marginal improvements especially considering the presence of noise".

---

> > > ### Author Response · Authors · 2021-11-27
> > > **A kind reminder**
> > >
> > > Dear Reviewer bgyC, thank you for your time to review our paper and engage in the constructive discussion. We are wondering whether you have got a chance to read our response? We will be glad to provide more explanations and answer more questions if you have any.

---

> ### Author Response · Authors · 2021-11-18
> **[Continued] Response to Reviewer bgyC**
>
> Q: Performance gain of CUMA is insignificant.
>
> A: Per your suggestion, we run our method over three different random seeds, and report the mean and standard deviation of the results in the updated PDF file. As shown in the updated Tables 1-5, our method outperforms all baselines by a considerable margin and the improvements are statistically reliable over different random seeds (below we show the mean and standard deviation of the improvements):
> - In Table 1, with uniform noise, CUMA outperforms the strongest baseline (LAFTR) by 5.09$\pm$0.82% on ∆EO.
> - In Table 2, with Gaussian noise, CUMA outperforms the strongest baseline (LAFTR) by 3.02$\pm$0.88% on ∆EOpp.
> - In Table 4, with Impulse noise, CUMA outperforms the strongest baseline (LAFTR) by 3.38$\pm$1.14% on ∆EO. With Gaussian Noise, CUMA outperforms the strongest baseline (LAFTR) by 3.11$\pm$1.28% on ∆EO.
> - In Table 5, with Impulse noise, CUMA outperforms the strongest baseline (AdvDebias) by 1.78$\pm$0.15% on ∆EO. With Gaussian Noise, CUMA outperforms the strongest baseline (AdvDebias) by 1.64$\pm$0.47% on ∆EO.
>
> Q: Compare with "FARF: A Fair and Adaptive Random Forests Classifier".
>
> A: Thanks for pointing out this interesting and relevant paper. Both FARF and our paper study the problem of fairness under distributional shifts. However, they have different problem settings and are not directly comparable.
>
> Specifically, as an online learning algorithm, FARF **adapts** the model to be fair on the current data distribution, from which training data is sampled for FARF (**seen** domain shifts). However, our work aims to **generalize** fairness learned on current distribution to **unknown** target distributions (**unseen** domain shifts). In our setting, the algorithm can not access any training data from the unknown target distributions.
>
> We will discuss the difference between FARF and our method, and highlight the importance of FARF to the community in our paper.

---

### Official Review · Reviewer_cgsp · 2021-11-01

**Correctness:** 2
**Technical Novelty And Significance:** 2
**Empirical Novelty And Significance:** 2
**Recommendation:** 3
**Confidence:** 4

**Main Review:**

Strength
  - The idea is new and interesting.
  - The paper is well written.

Weakness
  - I do not understand the fundamental claim that the similarity of the distributions of local curvatures
  ensure the fairness robustness. Even there is no explicit definition of fairness robust to distribution shift.

- I agree that local curvatures affects the robustness of a prediction model but I do not understand why and how  local curvatures affect the degree of fairness robustness.

- Numerical results do not  seem to support the main claim. For standard fair learning algorithms (not
robust to distribution shift) including AdvDebias and LAFTR, the degree of unfairness (e.g. $\Delta E_{OPP}$) even decreases as noises are added in the test data.

- Adding a small noise would not be sufficient to see the effect of robustness. As  (Ensuring Fairness Beyond the Training Data, NIPS2020) is done, the worst case distribution shift could be considered for numerical comparison.

**Summary Of The Paper:**

The paper proposed a new penalty term which is expected to make the final prediction model
robust to distribution shift of test data. The proposed penalty is based on the
distributions of local curvatures of two sensitive groups and enforces these two distributions to be similar.
The proposed idea is motivated by robust learning methods where the prediction model is robust to distributional shift.

**Summary Of The Review:**

I do not understand why making the prediction model robust results in fairness robustness.
For this reason, I recommend "rejection". However, it the authors explain successively why and how the two concepts - prediction robustness and fairness robustness, are related and how the proposed method utilizes this relation. Instead of intuitive explanations, some equations with rigorous definitions
would be helpful for understanding the value of the proposed method.

---

> ### Author Response · Authors · 2021-11-18
> **Response to Reviewer cgsp**
>
> We appreciate your frankness in expressing your difficulty in comprehending our core concepts. That is truly important for us to engage in an effective discussion. Precisely as you said, we agree that both intuitive explanations and equations with rigorous definitions are necessary in explaining our ideas. Actually, both math definitions and concrete examples have already been provided in the paper, if you could read again. For your convenience, we re-explained those concepts to you with both math definitions and concrete examples, and hope they clarify all your confusions and change your rating impression positively.
>
> Q: The definition of fair robustness.
>
> A: The definition of fair/equalized robustness is that we want the model to be equally robust to distributional shifts on two different subgroups (e.g., male and female) of data. For example, the model should have similar average accuracies on male and female test data with distributional shifts from the training set. In our experiments, we simulate such distributional shifts by adding noises on the original test data. (We also validated the effectiveness of our method under realistic domain shifts caused by time: See our replies to Reviewer ebvu’s 3rd question for details.)
>
> A more detailed description has already been provided in the original paper. See the paragraph under Figure 1 (in page 2): We first propose a new fairness objective, termed Equalized Robustness (ER), which aims to impose “equalized robustness” against unseen distribution shifts across the majority and minority groups, so that the fairness learned on in-distribution data can sustain on out-of-distribution test data. ER explicitly considers a new dimension of fairness that is practically significant yet so far largely overlooked. In other words, ER assesses the fairness on “out-of-distribution”, which has never been examined before and is found to be indeed fragile by our experiments. Therefore it works as a complement instead of a replacement for previous fairness metrics, which focus on assessing the “in-distribution” fairness.
>
> We also rigorously defined a new fairness metric ∆ER to measure the level of fairness in robustness (quoted from page 4): “Consider a machine learning model f trained on two groups of data $X_1$ and $X_2$ respectively. Suppose $C(X_1) ∼ P_1$ and $C(X_2) ∼ P_2$, where $C$ is the curvature function defined in equation (8), then the model’s ∆ER is defined as the squared maximum-mean-discrepancy (MMD) distance between $C(X_1)$ and $C(X_2)$: ∆ER = $MMD^2(P_1; P_2)$”. The smaller ∆ER, the smaller distance between the curvature distributions of two groups, and the better fairness in robustness.
>
>
> Q: Why and how local curvatures affect the degree of fair robustness.
>
> A: The short answer is: Curvature around a sample x affects prediction robustness on x against input perturbations. Optimizing the mean curvature over all training samples (as done in traditional robust learning papers) only results in robustness instead of robust fairness. The similarity/distance between curvature distributions of two groups of data (e.g., male and female) reflect fairness of robustness. Our method (CUMA) minimizes the distance/maximizes the similarities between the two curvature distributions of majority and minority groups, and achieves robust fairness.
>
> A more detailed explanation:
> Local curvature around a data point x is highly correlated with prediction robustness on $x$: if the curvature is small (i.e, the model is smooth around input data $x$), then the prediction on $x$ is robust to input perturbations. If the mean local curvature on the two groups (male and female) data are similar, then the mean robustness of the two groups are similar, and better fair/equalized robustness is achieved between that majority (e.g., male) and minority (e.g., female) data. In practice, we not only require the mean curvature (the first order momentum of the curvature distribution) of two groups to be similar, but we use a stronger requirement that the entire distribution of local curvature of two groups should be similar.
>
> The above explanations are formally summarized in the definition of “our new fairness metric ∆ER” in page 4. We use ∆ER (see definition in the answer to the above question or Eq. (5) in the paper) to measure the degree of fair robustness: The smaller ∆ER, the smaller distance between the curvature distributions of two groups, and the better fairness in robustness. With that said, we will add the above explanations to the paper to make it more clear.

---

> > ### Comment · Reviewer_cgsp · 2021-11-21
> > **Further questions**
> >
> > I still do not understand why fair robustness improve the standard fairness measure such s DP and EO.
> > Consider DP. Suppose that the score function ( even the authors called a continuous function as a classifier)
> > is given as $f(x)=1.0$ for man and $f(x)=-1.0$ for woman. Further, suppose that the domain of $x$ for man and domain
> > of $x$ for woman are disjoint. In this case, the curvatures of the score function for man and woman  are all 0 and hence
> > they are perfectly fair robustness but the DP is large.
> >
> > The definitions of Equalized Odds and Equalized Opportunity are not standard. As far as I know, it is
> > $P(C=1|A=0,Y=1)=P(C=1|A=1,Y=1)$ or something like that. I do not understand why the authors do not investigate the DP in their numerical works.

---

> > > ### Author Response · Authors · 2021-11-21
> > > **Further explanations**
> > >
> > > Thank you for your response! We apologize for the confusion caused by our writing and provide further explanations below.
> > >
> > > Q: Why fair robustness improve the standard fairness measure such as DP and EO?
> > >
> > > A: The short answer is: standard/in-distribution fairness is not a result of fair robustness (i.e., robust fairness as the term used in our paper). They are two different goals which are **simultaneously** optimized by our method.
> > >
> > > There are two fairness goals in our framework: the traditional in-distribution fairness and the new robust fairness.
> > > We **simultaneously** encourage both, instead of only encouraging robust fairness.
> > > This is achieved by jointly optimizing three loss terms in a **minimax** optimization framework in our final loss function (Eq. (10) on page 6):
> > > - $\mathcal{L}_{clf}$ encourages the model to give correct predictions (i.e., to encourage prediction accuracy).
> > > - $\mathcal{L}adv$ is the fairness loss term used in previous fair learning algorithms, such as AdvDebias and LAFTR. Used together with the above $\mathcal{L}_{clf}$ in a minimax optimization, it basically encourages correct predictions to be made (on in-distribution data) without utilizing the sensitive attribute $A$ as a feature (i.e., to encourage in-distribution fairness).
> > > - $\mathcal{L}_{cm}$ is our new fairness loss term to match the curvatures of two groups of data (i.e., to encourage robust fairness).
> > >
> > > As a result, our method achieves a good trade-off point among three dimensions: accuracy, in-distribution fairness, and robust fairness.
> > > In contrast, previous fair learning algorithms, such as AdvDebias and LAFTR, only consider the trade-off between accuracy and in-distribution fairness, and empirically achieve poor robust fairness according to our experimental results.
> > >
> > > So, in your example, such score function $f$ indeed has perfect curvature matching loss (i.e., $\mathcal{L}_{cm}=0$), **but it violates the regularization provided by $\mathcal{L}adv$**. As a result, such score function $f$ provided in your example is not a good solution for our optimization goal (Eq. (10) on page 6).
> > >
> > > In summary, our framework (as illustrated in Figure 2) and optimization goal (Eq. (10)) requires the model to **simultaneously** have good in-distribution and robust fairness.
> > >
> > > Also, our intuition to jointly optimize the two fairness goals is illustrated in Figure 1, if you could kindly check it again. It intuitively shows the difference between our "joint in-distribution and robust fairness" goal (Figure 1 (c)) and the "only in-distributional fairness" goal of traditional fair learning (Figure 1 (b)).
> > >
> > > We will add these explanations to our paper, which we believe will make it more accessible.

---

> > > ### Author Response · Authors · 2021-11-21
> > > **[Continued] Further explanations**
> > >
> > > Q: The definitions of Equalized Odds and Equalized Opportunity are not standard.
> > >
> > > A: The definitions in our paper follow [3] (see Section 2.1 in [3]), which are **equivalent** as the ones in the original EO paper [1] (and also the one you provided). Below we give derivations. Following your notation, we use C as the prediction variable (i.e., $\hat{Y}$ in our paper).
> > >
> > > In the original EO paper [1], the definition of EO is:
> > > P(C=1|A=0,Y=y) = P(C=1|A=1,Y=y), $\forall y \in {0,1}$ (1)
> > >
> > > In our paper, the definition of EO is:
> > > P(C$\neq$Y|A=0,Y=y) = P(C$\neq$Y|A=1,Y=y), $\forall y \in {0,1}$ (2)
> > >
> > > Below we derive the original definition (i.e., Eq (1)) from our definition (i.e., Eq (2)).
> > >
> > > When $y=1$, Eq (2) is P(C$\neq$Y|A=0,Y=1) = P(C$\neq$Y|A=1,Y=1),
> > >
> > > which is equivalent with P(C$\neq$1|A=0,Y=1) = P(C$\neq$1|A=1,Y=1), since the probabilities are already conditioned on $Y=1$.
> > >
> > > If we multiply both sides with -1 and the add 1 to both sides, we get: 1-P(C$\neq$1|A=0,Y=1) = 1-P(C$\neq$1|A=1,Y=1),
> > >
> > > which is equivalent with Eq (1) with $y=1$: P(C=1|A=0,Y=1) = P(C=1|A=1,Y=1). This is also the definition you suggested.
> > >
> > > When $y=0$, Eq (2) is P(C$\neq$Y|A=0,Y=0) = P(C$\neq$Y|A=1,Y=0),
> > >
> > > which is equivalent with P(C$\neq$0|A=0,Y=0) = P(C$\neq$0|A=1,Y=0), since the probabilities are already conditioned on $Y=0$.
> > >
> > > This is simply P(C=1|A=0,Y=0) = P(C=1|A=1,Y=0),
> > >
> > > which is equivalent with Eq (1) with $y=0$: P(C=1|A=0,Y=0) = P(C=1|A=1,Y=0).
> > >
> > > Q: Why not investigate the DP in numerical works?
> > >
> > > A: As pointed out by the paper of EO and EOpp [1] and even earlier in [2], DP (i.e., P(C=1|A=0)=P(C=1|A=1)) has two noticeable flaws:
> > > - First, DP doesn’t ensure fairness, since it doesn't take the ground truth Y into consideration. Consider a model for deciding the qualified students for college admission, where C=1 means the student is predicted to be qualified, Y=1 means one is actually qualified, and A is the sensitive attribute such as sex. DP allows the college to admit students who don't qualify (i.e., give predictions C=1 on the samples with ground truth Y=0), while rejecting those who actually qualifies (i.e., give predictions C=0 on the samples with ground truth Y=1), as long as the acceptance ratio of the two groups (i.e., P(C=1|A=0) and P(C=1|A=1)) are identical. This is unfair for those who actually qualifies.
> > > - Second, DP cripples with model utility (e.g., prediction accuracy). For example, the ideal predictor C=Y has prefect utility (i.e., prediction accuracy). If A is correlated with Y (which is common), then P(Y|A=0)$\neq$P(Y|A=1), and the ideal predictor violates DP condition: P(C|A=0)$\neq$P(C|A=1) where C=Y. In contrast, the ideal predictor (i.e., C=Y) satisfies EO, since P(C=1|A=0,Y=1) = 1 = P(C=1|A=1,Y=1) and P(C=1|A=0,Y=0) = 0 = P(C=1|A=1,Y=0).
> > >
> > > Please check Section 1 in [1] for more details on the limitations of DP.
> > >
> > > Following the suggestions in [1] and also the practical routines in recent papers [4,5], we use the more advanced metrics EO and EOpp, instead of DP,  for fairness evaluation.
> > >
> > > [1] Equality of Opportunity in Supervised Learning. NeurIPS, 2016.
> > >
> > > [2] Fairness Through Awareness. ACM ITCS, 2012.
> > >
> > > [3] Learning adversarially fair and transferable representations. ICML, 2018.
> > >
> > > [4] Fair Feature Distillation for Visual Recognition. CVPR, 2021.
> > >
> > > [5] Learning Disentangled Representation for Fair Facial Attribute Classification via Fairness-aware Information Alignment. AAAI, 2021.

---

> > > ### Author Response · Authors · 2021-11-23
> > > **A kind reminder**
> > >
> > > Dear Reviewer cgsp, thank you for your time to review our paper and engage in the constructive discussion. This is just a kind reminder that the close date of discussion is approaching. We are wondering whether you have got a chance to read our response to your follow-up questions? We will be glad to provide more explanations and answer more questions if you have any.

---

> > > ### Author Response · Authors · 2021-11-28
> > > **Humble reminder: any further question before you could re-assess our work?**
> > >
> > > Dear Reviewer cgsp,
> > >
> > > We are very thankful for you actively engaging in a constructive clarification/discussion with us!
> > >
> > > Since the discussion period is approaching its end, we wonder if you have any further question. At this moment, from our end, we feel we should have clarified all concerns and misunderstandings from your raised points.
> > >
> > > **If you could please check and let us know if you might be able to consider our work more positive, then we owe many thanks! Otherwise, if you have any further questions, please raise them and we're ready to clarify further any time.**
> > >
> > > With respects
> > > Authors

---

> ### Author Response · Authors · 2021-11-18
> **[Continued] Response to Reviewer cgsp**
>
> Q: For standard fair learning algorithms (not robust to distribution shift) including AdvDebias and LAFTR, the degree of unfairness (e.g. ΔEOpp) even decreases as noises are added in the test data.
>
> A: There is no guarantee on how the degree of unfairness (e.g., ΔEOpp) changes after adding noises in the test set for standard fair learning algorithms. In many cases, the degree of unfairness increases after the distribution shift. For example, in Table 1, ΔEOpp increases by 3.37% on LAFTR after adding uniform noise. In Table 4, ΔEOpp increases by 2.54% on AdvDebias after adding Gaussian noise.
>
> Also, as shown in the paper, our method achieves the best fairness under distributional shifts.
>
>
> Q: Experiments on worst-case distribution shift.
>
> A: We report the ΔEOpp (smaller is better) on the worst-case distribution shift on C&C dataset with “RacePctBlack” as the sensitive attribute.
>
> Normal: 39.64%
>
> AdvDebias: 27.86%
>
> LAFTR: 22.52%
>
> CUMA: 15.41%
>
> The results show that CUMA outperforms all baseline methods in terms of fairness under the worst-case distribution shift.

---

### Official Review · Reviewer_ebvu · 2021-11-02

**Correctness:** 3
**Technical Novelty And Significance:** 3
**Empirical Novelty And Significance:** 3
**Recommendation:** 6
**Confidence:** 2

**Main Review:**

Comments:
* Although the ideal model would indeed be robust against any unforeseen distributional shift, in certain applications (e.g., demographic changes in a student population applying for college), these changes are less dramatic over time (altough not necessarily predefined as described on page 4, Section 3). Furthermore, the decision-maker (e.g., school) might have some external information about the type of distributional shift. How can we get better guarantees in this case? Would the algorithm still be applicable with some modification to take into account this knowledge (and how)?
* In terms of writing, the paper is well written and precise but remains less accessible to readers who are familiar with the fairness literature but not other related parts (e.g., robustness and smoothness).
* In particular, this makes it harder for me to evaluate the novelty of the methodological contribution. Does the algorithm bring new ideas (of potential general interest) or just adapt existing robustness techniques to this particular fairness-related application? If it is the latter, what potential difficulties and special conditions one needs to address (that make the problem challenging)?
* I found the experiments extensive enough. However, a major question I have is how the distributional shift is modeled in those static datasets? On page 7, the paper mentions that Gaussian and uniform noise is used. First, I think that these shifts are not necessarily representable of the true shifts in a population (if not, are there some real cases that justify this assumption?). Second, this approach essentially introduces a “synthetic” distributional shift. It think it would make more sense to perform the same experiments on a dataset with observations over multiple years or after a policy change. The idea is to test how a model trained on old data performs on the most recent ones. I suspect that the changes might be not be very dramatic (since we are talking mostly about demographic changes that are slower).
* Given the focus on fairness, the choice of the CelebA dataset (with sensitive features “eyeglasses” and “chubby”) is odd and this case study could potentially be omitted.

Minor comments:
* the bright green color in links is difficult for reading
* I am not very familiar with the ICLR format, but Section 2 reads like a mix of math preliminaries and related literature. I think it’s better to shorten and remove unnecessary mathematical definitions.
* Typo: “the the stability”, p.1



**Summary Of The Paper:**

The paper is motivated by a common problem in real world applications of deep models, distributional shifts, which can cause unreliable behavior in the deployed models. In particular,  state-of-the-art fairness algorithms would be affected by such distributional shifts in the test data. This poses the following question studied by the paper: how can one achieve fairness when there exist unseen distributional shifts?

Towards this end, the paper proposes a new objective: Equalized Robustness (ER), that imposes equalized model robustness against distributions shifts across majority and minority. Fruthermore, the paper develops a new algorithm called Curvature Matching (CUMA) that imposes ER during training, and tests it through experiments.



**Summary Of The Review:**

The paper tackles an emerging problem (fairness under distributional shifts) and proposes a new objective and algorithm. While both the metric and the algorithm seem novel, I am not entirely convinced about the methodological contribution overall, given that I am not familiar with the general literature on robustness against distributional shifts. I also have concerns about the value of the experimental evaluation as explained in my comments above.

---

> ### Author Response · Authors · 2021-11-18
> **Response to Reviewer ebvu**
>
> Thanks for acknowledging the importance of the problem we studied and the novelty of our new fairness metric. Below we provide detailed answers to your questions. We hope they address your concerns and make you more convinced about the contributions of our work.
>
> Q: How to take into account prior knowledge about distributional shifts.
>
> A: We can modify the definition of curvature in Eq. (4) to utilize that prior information on distributional shifts.
>
> For example, we have three features in each sample: age, weight and sex. For simplicity, let's denote them as x1 (age), x2 (weight) and x3 (sex). And a data sample x=[x1,x2,x3] is a feature vector with three elements. The current method ensures the model to be fair under distributional shifts on all features. If we know that the distributonal shift is caused by changes in age and weight, while the sex attribute in the group remains unchanged, then we can redefine the curvature in Eq. (4) by replacing x with [x1,x2]. The modified method essentially ensures the model to be fair under distributional shifts on x1 (age) and x2 (weight), but not x3 (sex). This is acceptable since the prior knowledge tells us there is no distributional shifts on x3 (sex). Using prior knowledge, the modified method should give better results then the default method. This is because we removed the redundant constraint on x3 (sex), so that the model can learn better fairness on the other two features.
>
>
> Q: Does the algorithm bring new ideas (of potential general interest) or just adapt existing robustness techniques to this particular fairness-related application? If it is the latter, what potential difficulties and special conditions one needs to address (that make the problem challenging)?
>
> A: We are inspired by the curvature in (Moosavi-Dezfooli et al., 2019) as the robustness metric in defining our new robust fairness metric. But such adaptation is not straightforward, with the challenges described below.
>
> First, traditional robust learning papers such as (Moosavi-Dezfooli et al., 2019) minimize the mean curvature over all samples. However, in unbalanced datasets with majority and minority groups, such traditional robust learning methods would result in unbalanced robustness: the majority group can have much better robustness than the minority group. In contrast, our method aims to achieve fair robustness, by minimizing the discrepancy between the curvature distributions on two groups (majority and minority) of data - an untackled angle by any existing robust learning paper.
>
> Second, as explained in the last paragraph on page 4, our “out-of-distribution” fairness learning is also different from traditional “in-distribution” fairness learning. The target random variable in traditional “in-distribution” fairness learning follows a Bernoulli distribution and the distance between two Bernoulli distributions can be directly calculated.
> In contrast, the local curvature used in our  “out-of-distribution” fairness learning is a continuous random variable without a simple underlying distribution. As a result, we cannot directly measure the “out-of-distribution” fairness using the divergence between the two distributions as done in traditional “in-distribution” fairness learning. Instead, we optimize the MMD between two curvature distributions to achieve equalized robustness.
>
> We will update the paper to make this part more accessible to readers unfamiliar with relative literatures.
>
> Q: Synthetic v.s. real distributional shifts.
>
> A: Yes, we add Gaussian and uniform noise on the original data to generate synthetic distributional shifts. This is a more widely used setting in computer vision datasets, where the test samples are likely to have visual corruptions due to digital noises, movement of the camera or bad weathers, as shown in (Hendrycks & Dietterich, 2019).
>
> We agree it is necessary to evaluate the method on real distributional shifts. We test the models trained on the Adult dataset (US Census data collected before 1996) on the 2014 subset of  Folktables dataset (US Census data collected in 2014). This is a practical setting to evaluate real-world distributional shifts caused by time. We use “sex” as the sensitive attribute and compare the ΔEOpp and ΔEO (the smaller the better) of different methods on Folktables (2014) subset:
>
> Normal: ΔEOpp=12.64%, ΔEO=22.58%
>
> AdvDebias: ΔEOpp=10.84%, ΔEO=13.65%
>
> LAFTR: ΔEOpp=9.84%, ΔEO=12.10%
>
> CUMA: ΔEOpp=6.21%, ΔEO=7.89%
>
> The results show that CUMA outperforms all baseline methods on fairness under real-world distributional shifts.
>
> Q: Removing CelebA dataset.
>
> A: CelebA is also a popular benchmark dataset for fairness learning (Creager et al., 2019). The privacy attributes we used on CelebA follows (Creager et al., 2019). The experiments on it show our method works not only for Tabular datasets (e.g., Adult, C&C) but also on image datasets.
>
> Q: Typos and coloring.
>
> A: Thanks for your suggestion and we will fix them.

---

### Author Response · Authors · 2021-11-29
**General response, and in particular regarding Reviewer bgyC's problematic comments**

Dear Reviewers and AC panel,

Thank you again for reviewing our paper and providing helpful comments! We are glad the merits of our paper are enthusiastically acknowledged by Reviewers ebvu and xUAG.

The authors are an experienced research team with many experiences at ICLR and similar venues. We fully understand the workload of reviewers and ACs, and that getting timely yet quality comments is often hard. However, as authors, we also feel obliged to ensure our responses are "heard", rather than treated with nothing but irresponsible silence, as also encouraged by ICLR AC policies this year. In this view,  **we have to argue here, that a significant portion of comments from Reviewer bgyC appears to be vague and factually biased**.

**We have strived to address Reviewer bgyC's concerns and invited him/her for further discussions. Unfortunately, we haven't received a single response in a ten-day-long waiting time window, since our response on Nov. 19, despite repeated reminders**. Given that, we would like to publicly point this out to the audience and to make the AC explicitly aware of this situation.  To give a quick case review, we summarize Reviewer bgyC's comments and our response points below.

**1. Compare with another baseline method termed FARF**

Reviewer bgyC insisted that the online learning paper named FARF can be applied in our problem setting and should be compared with. We are happy to cite this paper, but we did not see how a comparison could be valid nor necessary between it and our method.

As an online learning algorithm, FARF **adapts** the model to be fair on the current data distribution, from which training data is sampled for FARF (**seen** domain shifts). However, our work aims to **generalize** fairness learned on current distribution to **unknown** target distributions (**unseen** domain shifts). In our setting, the algorithm cannot access any training data from the unknown target distributions. It is not applicable for FARF to adapt the model to an unknown target distribution, from which one cannot sample any training data.

We are hence confused why keeping asking for our comparison with this one:

"FARF: A Fair and Adaptive Random Forests Classifier", PAKDD 2021
- Author list: Wenbin Zhang, Albert Bifet, Xiangliang Zhang, Jeremy C. Weiss, Wolfgang Nejdl
- arXiv link: https://arxiv.org/abs/2108.07403

We welcome any further discussion.

**2. Limited novelty**

Also, Reviewer bgyC vaguely accused "limited novelty", which seems to also arise from his/her strong (mis)understanding that our task setting is similar to that in FARF. Respectfully yet firmly, we believe his/her understanding to be wrong as aforementioned explained.

Besides, all other three reviewers acknowledged the novelty of our new fairness goal:

Reviewer ebvu: “The paper proposes a new objective: Equalized Robustness (ER), that imposes equalized model robustness against distributions shifts across majority and minority.”

Reviewer cgsp: “The idea is new and interesting.”

Reviewer xUAG: “This paper proposes a new fairness metric, Equalized Robustness (ER), to assess model robustness for sensitive subgroups.”


**3. "Marginal Improvement" and "Noise and uncertainty" in results**

Reviewer bgyC accused that our method makes "marginal improvements" and has "noise" and uncertainty in results.

With all respect, we think his/her claim is not well supported. The main results of our method are all averaged over three random runs, then outperforming all baselines by a considerable margin (below we show the mean and standard deviation of the improvements):

In Table 1, with uniform noise, CUMA outperforms the strongest baseline (LAFTR) by 5.09$\pm$0.82% on ∆EO.

In Table 2, with Gaussian noise, CUMA outperforms the strongest baseline (LAFTR) by 3.02$\pm$0.88% on ∆EOpp.

In Table 4, with Impulse noise, CUMA outperforms the strongest baseline (LAFTR) by 3.38$\pm$1.14% on ∆EO. With Gaussian Noise, CUMA outperforms the strongest baseline (LAFTR) by 3.11$\pm$1.28% on ∆EO.

In Table 5, with Impulse noise, CUMA outperforms the strongest baseline (AdvDebias) by 1.78$\pm$0.15% on ∆EO. With Gaussian Noise, CUMA outperforms the strongest baseline (AdvDebias) by 1.64$\pm$0.47% on ∆EO.

This shows the improvements made by our algorithm are **large** and **statistically significant**. We truly have no idea how Reviewer bgyC draws his/her conclusion on this one.

In summary, we find none of the negative reasons given by Reviewer bgyC is grounded on truth. We are surprised that this reviewer won't engage in any further discussion after making accusations. Therefore, we have to draw the attention of AC and the public audience so as not to be misled.

We should conclude by stating that our arguments are NOTHING PERSONAL (we sincerely respect reviewers), yet only restricted to a fact-based discussion of this current case.

**Given the above, we politely ask AC and other reviewers to disregard those problematic comments.**

Best

Authors

---

### Decision · Program_Chairs · 2022-01-20

**Decision:**

Reject

**Comment:**

The paper considers learning a fair classifier under distribution shift. The proposal involves an additional MMD penalty between the model curvatures on the data subgroups defined by the sensitive attribute. Reviewers generally found the problem setting to be well motivated, and the paper to have interesting ideas. Some concerns were raised in the initial set of reviews:

(1) _Relation between local curvatures and fairness robustness_. The concern was that the paper does not make sufficiently clear how similarity of the distributions of local curvatures ensures fairness robustness, and that there is no explicit definition of fairness robust to distribution shift.

(2) _Comparison to related work_. The concern was that works such as FARF as also considering the issue of distribution shift.

(3) _Technical novelty_. The concern was that technical depth of the proposal may be limited, as it builds on existing ideas (e.g., adversarial learning, Hessian to measure curvature).

(4) _Significance of results_. The concern was that the improvements of the proposed method are not significant, statistically and/or practically.

For point (1), the response clarified that the proposal is to ensure that the local curvature (and hence robustness) across data subgroups is similar. The relevant reviewer was still unclear as to whether this ensures what one might intuitively consider "robust fairness". On my review of the paper, I do concur that from the Introduction, and the para preceding Eqn 4, it appears that one natural notion is

$ \sup_{\mathbb{Q} \in \mathcal{U}( \mathbb{P} )} \Delta( \mathbb{Q}( \hat{Y}, Y \mid A = 0 ), \mathbb{Q}( \hat{Y}, Y \mid A = 1 ) ) $

where $\mathbb{P}$ is the observed data distribution, $\mathcal{U}$ is some uncertainty set, and $\Delta$ is some fairness measure (e.g., DP). Assuming this is indeed the ideal, it would be useful to mathematically contrast it to the proposal adopted in the present paper. The para preceding Eqn 4 correctly notes that the above notion would require specifying $\mathcal{U}$. This may be challenging, but an apparently reasonable strategy that follows the distributionally-robust optimization literature would be to use a specific ball around the training distribution (e.g., all distributions with bounded KL divergence against $\mathbb{P}$). Further, it is of interest to ask whether the proposed objective in any way approximate this one; put another way, is there any implicit assumption made as to which class of distributions one is likely to encounter?

Further discussion would also be useful on the following alternative to the objective presented in the paper: rather than match the curvatures for the subgroups, simply minimise their unweighted average. This ought also to ensure robustness under the two different distributions; page 2 hints that this might not work owing to the different scales of these terms (i.e., the minority subgroup being much less robust), but the point does not seem to be discussed very explicitly subsequently.

For point (2), the response noted that FARF is designed for online learning, whereas the present paper involves a single, static training set drawn iid from a single distribution. In the present paper, the drift happens at test time, and the learner has no access to samples from this distribution. The authors argued that FARF can be applied as-is to this setting. From my reading of this and the FARF paper, I agree that while the latter should be cited, it is not clearly applicable to the present setting.

This said, the present paper primarily focusses on the covariate shift setting, for which there have been some relevant recent works; see:

Singh et al., "Fairness Violations and Mitigation under Covariate Shift", FAccT '21.

Rezaei et al., "Robust Fairness under Covariate Shift", AAAI '21.

The former uses tools from joint causal graphs, while the latter assumes access to an unlabelled sample for the target distribution. The present work is certainly different in technical details, but at a minimum it seems prudent to acknowledge that there are relevant works on ensuring fairness outside the observed training distribution, and thus tone down statements such as "As a pioneer work...". There also seems scope to compare against the latter, e.g., to see how valuable having a few samples from the target domain are.

Another work relevant to the spirit of ensuring fairness beyond the observed data is

Mandal et al., "Ensuring Fairness Beyond the Training Data", NeurIPS 2020.

This is in line with the distributionally-robust objective suggested in point (1), where one considers test distributions that can be arbitrary re-weightings of the training distribution.

For point (3), from my reading, the technical content is reasonable. I would however have liked more mathematical discussions on point (1) above, which is important as it is the foundation of the strategy followed.

For point (4), the response asserts their improvements are significant practically and statistically. From my reading, I am inclined to agree with this claim. I would however note that another reviewer raised the question of whether Gaussian and uniform noise are reflective of real-world distribution shifts. I concur with this concern; this part of the paper seems a little disappointing. The response mentioned results on a new setting with more realistic shift, which we suggest is incorporated into future versions of the paper.

Overall, the paper has some interesting ideas for a topical and important problem. At the same time, there is scope for tightening the work per the comments above, particularly on points (1) and (2), and to some extent (4). We believe that addressing these would help properly situate the work, and thus increase its clarity and potential impact. We thus encourage the authors to consider incorporating these for a future submission.